# Pollution and Cleaning of PDMS Pervaporation Membranes after Recovering Ethyl Acetate from Aqueous Saline Solutions

**DOI:** 10.3390/membranes12040404

**Published:** 2022-04-06

**Authors:** Xuefei Sun, Yang Pan, Chunxiang Shen, Chengye Zuo, Xiaobin Ding, Gongping Liu, Weihong Xing, Wanqin Jin

**Affiliations:** 1State Key Laboratory of Materials-Oriented Chemical Engineering, College of Chemical Engineering, Nanjing Tech University, 30 Puzhou Road(S), Nanjing 211816, China; sunxuefei@jiusi.com (X.S.); panyang@njtech.edu.cn (Y.P.); shenchunxiang@jiusi.com (C.S.); zuochengye@njtech.edu.cn (C.Z.); gpliu@njtech.edu.cn (G.L.); 2NanJing Jiusi High-Tech Co., Ltd., 1 Yuansi Road, Nanjing 211000, China; dingxiaobin@jiusi.com

**Keywords:** ethyl acetate, salt, wastewater, pervaporation, backflushing

## Abstract

The removal of volatile organic compounds (VOCs) from wastewater containing nonvolatile salts has become an important and interesting case of the application of the pervaporation (PV) process. The aim of this study was to evaluate the influence of salts on the PV removal of ethyl acetate from wastewater using a polydimethylsiloxane (PDMS) membrane. The fouled membrane was then characterized via scanning electron microscopy–energy-dispersive X-ray analysis (SEM–EDX) to investigate salt permeation. The membrane backflushing process was carried out by periodically flushing the permeate side of the tubular membrane. The results demonstrated that salts (NaCl and CaCl_2_) could permeate through the PDMS membrane and were deposited on the permeate side. The presence of salts in the feed solution caused a slight increase in the membrane selectivity and a decrease in the permeate flux. The flux decreased with increasing salt concentration, and a notable effect occurred at higher feed-salt concentrations. A permeate flux of up to 98.3% of the original flux was recovered when the permeation time and backflushing duration were 30 and 5 min, respectively, indicating that the effect of salt deposition on flux reduction could be mitigated. Real, organic, saline wastewater was treated in a pilot plant, which further verified the feasibility of wastewater PV treatment.

## 1. Introduction

Volatile organic compounds (VOCs), such as ethyl acetate, are present in chemical effluents and are considered harmful due to their impact on the environment [1]. These pollutants, originating from wastewater units, easily enter the air and indirectly impact human health [2,3]. Furthermore, VOCs play a critical role in the formation of tropospheric ozone [4,5]. In recent years, most chemical plants have employed affiliated facilities to purify wastewater before discharging it into the environment. Traditional treatment methods include sorption, extraction, and distillation, which exhibit certain disadvantages, such as a high follow-up treatment costs, large by-product volumes, and complex feed conditions [6,7,8]. Therefore, a highly efficient, cost-effective VOC recovery method is urgently needed. The development of methods to recover and reuse valuable compounds retrieved from chemical effluents has attracted growing interest.

Pervaporation (PV) is a processing method used to achieve the separation of mixtures of liquids via partial vaporization through a nonporous, permselective membrane [9]. The separation process of components is based on the difference in transport rates of individual components through a given membrane and can be explained by the solution–diffusion model [10]. Mass transfer across a PV membrane can be described by three major steps: (1) sorption of valuable components on the active surface layer of the membrane, (2) transportation of components across the membrane via diffusion, and (3) desorption on the permeate side into the vapor phase under vacuum [11]. The advantages over traditional methods include low energy consumption, high product purity, no environmental pollution, and simple operation [12]. In addition, various feed mixtures can be treated in one unit to realize a high degree of flexibility.

However, inorganic salts are also usually present in wastewater containing organic pollutants, limiting the application of PV membranes. Nonvolatile compounds can notably influence the PV performance. Salts affect both the mass transfer rate and driving force of the transport process of organic compounds through a polydimethylsiloxane (PDMS) membrane [13,14,15,16]. However, whether salts permeate through PV membranes and are deposited on the permeate side is a controversial subject. Lipnizki et al. examined the effects of NaCl, MgCl_2_, and glucose on the permeation performance of a 1-propanol/water mixture through commercial PDMS membranes PERVAP1060 and 1070. They found that NaCl penetrated the membrane structure and was present on the permeate side of the membrane [17]. Ravindra et al. reported that salt was deposited on the permeate side when processing liquid propellant reaction liquors at a NaCl concentration higher than 10 wt.% with a dense chitosan membrane. Higher salt concentrations in the feed affected both the total flux and salt rejection [18]. Furthermore, Zwijneneberg et al. described the separation performance and sensitivity towards the fouling of a new PV tubular membrane in a simulated, solar-driven process. They further reported the occurrence of salts in the permeate at the μg/L level [19].

In contrast, other researchers have reported that PV membranes remain impermeable to salts. Veronica et al. investigated the applicability of the PV process in the recovery of *n*-butanol from salt-containing mixtures. Their results demonstrated that neither Celfa nor P500-1 membranes accomplished permeation towards NaCl [16]. In addition, Xie et al. evaluated the effect of operating conditions on the PV-separation performance of aqueous salt solutions. Under all operating conditions, salt rejection reached up to 99.9%, indicating that the salt rejection performance of poly (vinyl alcohol) (PVA)/maleic acid (MA)/silica hybrid membranes was independent of operating conditions due to the nonvolatile nature of NaCl [20]. The significant controversy between the different studies concerning salt permeation may be attributed to material properties or the crosslinking degree of the membranes.

Based on the contradictory results described above, in this study, a series of tests was conducted to assess the permeability of PDMS membranes by permeating ethyl acetate/water/salt mixtures. To minimize salt deposition on membrane inner surfaces and thus maintain high permeate fluxes during operation, membrane cleaning was performed to improve the efficiency. The backflushing duration and frequency were varied at fixed crossflow velocity and temperature values to determine the best backflushing conditions that maintained the maximum flux without worsening the permeate quality. To further study the pollution and cleaning behavior of PDMS PV membranes, real, organic, saline wastewater was treated in a pilot plant.

## 2. Materials and Method

### 2.1. Materials

Two commercial PV membranes, tubular ceramic and flat-sheet composite membranes, provided by JiuSi High-Tech (Nanjing, China), were investigated in this study. Both membranes contain a selective PDMS layer with a thickness of 8 µm. The tubular ceramic composite membrane includes a porous ceramic support, while the flat-sheet composite membrane contains a polyetherimide (PEI) support. The characteristics of these membranes are summarized in Table 1. Scanning electron microscopy (SEM) images of virgin tubular PDMS/Al_2_O_3_/ZrO_2_ and flat-sheet PDMS/PEI composite membranes are shown in Figure 1. Ethyl acetate, NaCl, and CaCl_2_ were purchased as analytical reagents from Shanghai Macklin Biochemical Co., Ltd., China. Deionized water was employed in all solution preparations.

### 2.2. Experimental Device

A diagram of the laboratory operation unit is shown in Figure 2. The laboratory installation comprises two units, namely, PV and backflushing systems. The PV system was equipped with seven main elements, i.e., a feed tank, feed pump, preheater, membrane module, condenser, vacuum pump, and permeate tank. The membranes adopted in the backflushing experiments were PDMS/ceramic tubular composite membranes. The PV experiments were carried out at a constant temperature of 40 °C and a permeate pressure of 5000 Pa. The ethyl acetate concentration in the feed was always maintained at a saturated level (approximately 4.8 wt.%) to preclude any effects arising from variations in the ethyl acetate concentration. Excessive ethyl acetate was added to the saturated EA/water solution, and when ethyl acetate in the solution penetrated through the membrane, the upper ethyl acetate would dissolve, maintaining a saturated concentration. To reduce the effect of concentration polarization, the volumetric flow rate of the feed pump was adjusted to 15 L/min, with a crossflow velocity of 0.2 m/s to obtain turbulent flow conditions. Permeate vapor was collected in a liquid nitrogen trap, and the cold trap was exchanged at specific time intervals with consecutive permeate collection.

A backflushing device, which removed any deposited salts on the permeate side and increased the permeate flux, was mounted at the permeate port of the membrane module. The basic elements of the backflushing subsystem included a backflushing tank and a low-pressure pump. After each set of salt-solution experiments, the inside wall of the tubular composite membrane was flushed with pure water to wash away deposited salts, and the permeate flux, which was defined as recovered flux, was again measured under the initial testing conditions. The difference in the permeate flux was adopted as a measure of the membrane’s backflushing efficiency.

The flow diagram of the pilot-scale experiment is similar with the content of Figure 2. The pilot-scale device was mainly composed of a feed tank, a circulation pump, a heating system, membrane modules, a refrigeration system, a vacuum pump, and a permeate tank. The rated flow-rate of circulation pump was 5 m^3^/h, which can be adjusted by a frequency converter. The heating system consisted of a heated water bath (total power, 8 kW) and a plate heat-exchanger I (heat-exchange area, 2 m^2^). The refrigeration system consisted of refrigerator (total power, 6 kW) and a plate heat-exchanger II (heat-exchange area, 8 m^2^). The cooling medium (ethylene glycol/water = 6:4) flows into the cold side, while permeate vapor flows into the hot side of the heat-exchanger II. Three PDMS ceramic membrane modules were applied. Each module was comprised of 31 ceramic membrane tubes with an effective area of 0.67 m^2^. The feed temperature was 40 °C, the cooling medium temperature was −15 °C, and the vacuum pressure was 5000 Pa. The volume of the feed solution in each batch was 30 L.

### 2.3. Experimental Procedures and Tests

This experiment consisted of two parts, part (1): the laboratory test, and part (2): the pilot-scale experiment. In the laboratory test, ethyl acetate/water/salts simulated solutions were used as the feed solutions. The pervaporation tests were carried out for one hour to reach the steady state. Then, the actual tests were implemented by using two parallel cold traps to collect the permeation compound intermittently. The permeate flux was recorded at a given time (30 min). For supplementation, the collection interval could be tuned according to the actual requirement, as shown in Figure 3 and Figure 4. The composition of the permeate was analyzed by using a gas chromatographer (GC-2014, Shimadzu, Japan). It was equipped with a thermal conductivity detector (TCD) and a Porapak Q packed column. Helium (He) was chosen as the carrier gas. The permeate conductivity was measured by adopting a conductivity meter (RS 232—METER 8306). The pervaporation experiment was repeated at least three times to ensure data reliability.

Then, the SEM and EDX were adopted to determine whether salt could penetrate through the PDMS membrane. The effect of salt types and concentrations on the separation performance of the PDMS/ceramic tubular membrane and PDMS/PEI flat membrane were also investigated. Finally, different operating conditions of backflushing were discussed to obtain the optimum parameter. The permeate flux (J) was obtained by weighing the permeate product collected in a cold trap (W) or permeate tank for a given time (t). To prevent confusion, the membrane flux after water flushing process is defined as “recovered flux”.

In the pilot-scale experiment, actual industrial wastewater was chosen as the feed for the pervaporation process. The pilot-scale experimental system mainly consisted of three membrane modules placed in series. The permeate of former module was performed as the feed for the next stage, leading to the enrichment of the ethyl acetate. The ethyl acetate concentration on the permeate side could easily exceed the saturated solubility, and stratification could ensue. The lower layer was a saturated ethyl acetate aqueous solution, while the upper layer was pure ethyl acetate, which could be collected with a separating funnel. Several batches of pollution and cleaning were processed. The first step was opening the valve, and the pump with low pressure (pressure < 0.01 MPa) was applied to inject water into the inside of membrane tube. The flushing process was maintained for 5 min to ensure it was totally cleaned. After each batch was completed, all the liquid on the feed side was discharged with compressed air (maximum pressure, 0.05 MPa), and the experiment was then repeated with new wastewater.

The solution–diffusion model is an accepted model for describing the PV process [21]. The total flux (J) and separation factor (β) of the permeating components are generally calculated to evaluate the separation performance of the membranes. The partial flux ($J_{i}$) of compound *i* through the membrane is calculated according to Equation (1) [22,23]:(1)$$J_{i}=\frac{\Delta mi}{A\Delta t}$$
where Δ*m_i_* is the mass of component *i* (g) collected over a given period Δt(h) and A is the membrane area (m^2^).

The separation factor (β) is calculated by adopting Equation (2) [24]:(2)$$\beta_{ij}=\frac{{\;y}_{i}{/y}_{j}}{{\;x}_{i}{/x}_{j}}$$
where y*_i_* and y*_j_* denote the weight or molar fraction of components *i* and *j*, respectively, in the permeate, and x*_i_* and x*_j_* are the weight or molar fraction of components *i* and *j*, respectively, in the feed.

## 3. Results

### 3.1. Influence of Salts on the PV Performance

#### 3.1.1. Pervaporation Performance of the PDMS Membrane in the Presence of Salt

PV experiments were performed using PDMS/ceramic tubular composite membranes. Figure 3 shows the influence of the NaCl concentration and operation time on the water flux in binary NaCl–water solutions. In the pre-experiment, we found that the water flux decreased obviously for 1 h when the salt was introduced into feed solution. To observe the change of water flux more clearly, we had to record the data for 5 min. The interval of recording data is acceptable because the system had reached a steady state after 1 h of pre-operation. The water flux in the absence of NaCl remained constant at 0.42 kg/m^2^ h over 50 min of permeation for both the tubular and flat-sheet membranes. However, the water flux decreased linearly from 0.42 to 0.28 kg/m^2^ h for the tubular membrane in the binary NaCl–water solution experiments, whereas the water flux decreased from 0.43 to 0.32 kg/m^2^ h for the flat-sheet membrane. This result suggested that the presence of salt in the feed imposed a direct effect on the flux of pure water. The presence of salt typically increases the density and viscosity of the feed solution, which results in a reduction in the water activity of the NaCl solution [25]. As a consequence, the transport rate of water molecules is lower. The permeation flux of the ceramic membrane decreased faster than the flat membrane. This could occur because the smaller pore size of the flat substrate (20 nm) had an inhibition effect on the permeation of the salt, compared to the tube ceramic substrate (200 nm).

Figure 4 shows a summary of the comparative results obtained when NaCl was added to an ethyl acetate/water solution. The collection interval is similar with Figure 3. In the absence of salt, the PDMS/ceramic tubular membrane attained a constant flux of 1.8 kg/m^2^ h with a separation factor of 12.6 during the investigated period of 50 min. However, in the case of 15 wt.% salt, the permeate flux of the PDMS membrane remained nearly constant during the first 20 min and then gradually decreased from 1.77 to 1.42 kg/m^2^ h from 20 to 50 min. As shown in Figure 4b, the decrease of EA dominated the decline of permeate flux. The change of water flux could be ignored. This result indicated that the resistance to mass transfer of EA was considerably higher in the presence of salt than in the absence of salt. The presence of salt slightly affected the separation factor, which marginally increased after 50 min of permeation.

The fouled PDMS/ceramic tubular membrane after the test is shown in Figure 5. Notably, white salt crystals are deposited at the outlet of the tubular membrane, which provides evidence that salts can permeate through the PDMS PV membranes. Although the permeate conductivity was very low, i.e., 4.1 μS/cm, the hydrated ions of salt could pass through the polymer matrix via the free volumes. Meanwhile, the salt may also be transported through the tiny defects existing in the selective layer. We found that salt transport through our PDMS composite membrane is very complicated and dependent on various feed conditions and membrane microstructures. It will be further studied in our future work.

#### 3.1.2. Scanning Electron Microscopy–Energy-Dispersive X-ray Analysis (SEM–EDX) Observation of the Virgin and Fouled Membranes

SEM–EDX analysis (Figure 6 and Figure 7) was performed on cross-sections of virgin (a, b, and c) and fouled tubular PDMS membranes (d, e, f, g, and h), respectively. It was clearly observed that Na and Cl were evenly distributed throughout the entire cross-section of the fouled tubular PDMS membrane, which indicated that NaCl could diffuse through the PDMS membrane. Silicon elements were distributed in the ZrO_2_ support, which suggests the pore penetration of the casting solution. 

The obtained results, as shown in Table 2 and Table 3, indicated that the virgin membrane cross-section mainly contained C (17.21 wt.%), O (43.15 wt.%), Al (22.02 wt.%), Si (2.78 wt.%), and Zr (14.81 wt.%), while the fouled membrane contained two elements, namely, Na (0.14 wt.%) and Cl (0.47 wt.%), which was attributed to NaCl permeation.

#### 3.1.3. Influence of NaCl Concentration on Pervaporation Performance

As shown in Figure 8, we further investigated the effect of NaCl concentration on separation performance in an ethyl acetate/water/NaCl ternary solution. The results of each experiment with different NaCl concentrations were recorded after running for 10 min. 

The water flux and ethyl acetate flux remained approximately constant at low salt concentrations (1–7 wt.%). However, both partial fluxes exhibited a declining trend when the salt concentration was higher than 7 wt.%. There are multiple reasons for the above reduction with increasing salt concentration. At higher salt concentrations, more serious concentration polarization occurs, and the diffusing salts seem to block channels that facilitate the entry of water or ethyl acetate molecules into the membrane, thus reducing the total flux. Additionally, with increasing salt content, more ionic clusters are formed, and the electrostatic interactions intensify, resulting in the activity coefficients of water and solvent being largely impacted, affecting the PV driving force, and causing a notable decline in the total flux [26]. In addition, with increasing salt content, the density and viscosity of the feed solution increase. As a result, the transport rate is lower.

The separation factor slightly increased from 12.5 to 13.5, with increasing the salt concentration from 1 to 15 wt.%. The reason for this effect is the reduction in water vapor pressure or activity coefficients due to salt. When a salt is dissolved in a mixed solution containing two volatile liquid components, the dissolved salt could affect the activities of these two volatile components through the formation of liquid-phase associations or complexes. Since the activity of ethyl acetate increases by adding NaCl to the solution, the solubility in the membrane is higher than that of water [27]. Consequently, the separation factor increases upon the addition of salt to the solution.

#### 3.1.4. Influence of Various Salt Species on the Pervaporation Performance

We further investigated the effect of salt types on the separation performance of PDMS/ceramic tubular membrane and PDMS/PEI flat membrane. PV was continuously conducted over 50 min at 40 °C with 15 wt.% NaCl, 15 wt.% CaCl_2_, and 4.8 wt.% ethyl acetate solutions.

The driving force of the PV process is the difference in partial pressure or activity of the transported component. The addition of salts to the feed solution alters both the partial pressure or activity on the feed side and the flux. As expected, the total flux continuously declined with the operation time for both the tubular and flat composite membranes in the different salt solutions. It is accepted that the membrane resistance increases with increasing membrane thickness, resulting in a decrease in the permeate flux. However, the resistance of PDMS composite membranes largely depends on the top layer. Therefore, it should be noted that, given the same thickness of the top layer (8 µm), the initial fluxes of these two composite membranes are nearly identical, as shown in Figure 9.

The total flux decline appeared to be more severe for the tubular composite membrane with increasing operation time. For instance, the total flux of the tubular composite membrane decreased from 1.8 to 1.42 kg/m^2^ h over 50 min of operation, while that of the flat composite membrane decreased from 1.8 to 1.63 kg/m^2^ h for the CaCl_2_ solutions. This can be explained by salt crystallization in the membrane support, which can increase the resistance to vapor permeation. Although the top layer thicknesses for both membranes were identical, the support of the tubular composite membrane was much thicker than that of the flat composite membrane.

The results in Figure 9 indicate very similar performance levels of the tubular and the flat composite membranes for the NaCl and CaCl_2_ solutions, indicating that the salt type (i.e., NaCl and CaCl_2_) slightly affects the water flux at a given feed-solution salinity.

### 3.2. Membrane Cleaning

As discussed above, there was a negative effect on the PV separation performance at high salt concentrations in the feed solution. The observed flux drop was attributed to the reduced water activity in the presence of salt. In addition, physical salt deposition in the support membrane pores caused a gradual increase in membrane resistance. Therefore, it is necessary to develop strategies that prevent or reduce salt deposition. Backflushing is a widely implemented pollutant removal approach in industrial processes. A series of filtration experiments was therefore carried out to investigate the effect of the crossflow velocity, backflushing water temperature, and backflushing frequency on the permeate flux behavior. The tubular membrane was used for membrane cleaning because its module structure was more suitable for reverse cleaning of the permeate side.

#### 3.2.1. Effect of the Crossflow Velocity and Temperature on Permeate Flux Recovery

Solutions with constant ethyl acetate and NaCl concentrations of 4.5 and 15 wt.%, respectively, were applied as feeds during the PV process. Velocities of 0.5, 1.0, 1.5, and 2.0 m/s were investigated at various temperatures. The effect of the backflushing velocity on the permeate flux is shown in Figure 10. It was found that the higher the backflushing velocity applied to the inner wall of the tubular membrane was, the higher the recovered flux would be. The increase in recovered flux can be explained by the enhanced shear stress on the membrane’s inner surface, which increases the dissolution rate of deposited salts and thus enhances the recovered flux. At lower backflushing velocities, even a small increase in the backflushing velocity increased the flux more than did a similar increase at higher backflushing velocities.

The effect of the backflushing water temperature on the recovered flux may be attributed not only to the effect of the temperature on the solubility of the deposited salts but also to the complex physical change likely occurring in the composite membrane as the temperature was varied.

In this process, the backflushing water temperature was varied from 30 °C to 60 °C. Figure 10 shows that the recovered flux was sensitive to changes in the backflushing water temperature. At a backflushing water temperature of 30 °C, the salt solution coefficient was lower, resulting in a lower recovered flux. However, as the backflushing temperature increased from 50 °C to 60 °C, the crossflow velocity did not result in a clear improvement since the recovered flux slightly increased at all crossflow velocities. This indicates that a temperature around 50 °C is sufficiently high to dissolve any deposited salts on the membrane’s inner surface, and a further increase in backflushing water temperature does not significantly influence the recovered flux.

#### 3.2.2. Effect of the Permeation Time t_p_ and Backflushing Duration t_b_

To investigate the influence of t_p_ and t_b_, three t_p_ values (30, 60, and 90 min) and two t_b_ values were considered. Moreover, tests were conducted with and without backflushing to quantify the membrane flux recovery level under different operating conditions. The feed solution contained 4.5 wt.% ethyl acetate, 15 wt.% NaCl, and water. The crossflow velocity and backflushing water temperature remained fixed at 2.0 m/s and 50 °C, respectively. The effects of t_p_ and t_b_ on the recovered flux and the separation factor are summarized in Table 4.

When the backflushing technique was used, regardless of the backflushing conditions, the recovered flux increased. After 30 min of PV without backflushing, a flux decline of 8.3% occurred for the PDMS composite membrane. As indicated in Table 2, the maximum value of the recovered flux, i.e., 1.78 L/m^2^ h, was attained for t_p_ = 30 min and t_b_ = 5 min. The obtained results suggested that, when backflushing was applied for t_b_ = 10 min, the recovered flux did not improve significantly. Hence, after 30 min of PV, the recovered flux remained nearly constant. This result demonstrates that a longer backflushing duration is unnecessary, and the deposited salts can be adequately dissolved when the backflushing duration is 5 min. The recovered flux was found to decrease with increasing t_p_. A probable explanation for this phenomenon is that the larger the t_p_ value, the more intensely salt deposition occurs on the inner surface of the tubular composite membrane.

### 3.3. Treatment of Organic, Saline Wastewater in a Pilot Plant

As shown in Figure 11, wastewater treated in the pilot-scale experiment was acquired from Shandong Lukang Pharmaceutical Factory, China. The main components of the wastewater are listed in Table 5. The NaCl content in the treated wastewater was much higher than that of the other salts, so this experiment primarily analyzed the influence of NaCl on the membrane’s separation performance. The preliminary wastewater treatment process generally involves aeration, that is, air is forced through the wastewater, which can volatilize ethyl acetate. This method not only pollutes the air environment but also wastes ethyl acetate. Therefore, the PV membrane method is an economic and environmentally friendly treatment method. The pilot-scale experiment was conducted at a constant temperature of 40 °C and a permeate pressure of 5000 Pa.

#### 3.3.1. Variation in the Main Components during the Pervaporation Process

In the pilot-scale experiment, the ethyl acetate concentration in the retentate decreased from 0.65 to 0.04 wt.%. A small fraction of salt could diffuse through the membrane, but the majority of salt clogged the PDMS membrane. Theoretically, the salt content on the residual side gradually increases. However, an increase was not observed during the PV process, which could be explained by the fact that salts penetrated the membrane. The salt content in the permeate remained undetectable, and similar to the small-scale experiment, permeated salts accumulated on the permeate side of the membrane and could not be transmitted to the permeate. During the whole PV process, the permeate flux (of the EA/water mixture) gradually decreased from 0.96 kg/m^2^ h to 0.48 kg/m^2^ h, mainly because both the ethyl acetate content in the residual solution and that diffusing through the membrane decreased (Table 6). In addition, salt accumulation on the permeate side was one of the reasons explaining the decline in the permeate flux.

#### 3.3.2. Pollution and Cleaning during the Pilot-Scale Process

The salt content in the pilot-scale experimental wastewater was much lower than that in the laboratory test, so we did not expect that membrane fouling would be as severe. Therefore, the cleaning process in the pilot-scale test was performed after every three batches of PV process. 

A total of seven batches occurred in this experiment. The data in Figure 12 indicate that the total flux in the first three batches gradually decreased as a result of the membrane fouling of each batch. After three batches, the total flux dropped to 86.7% of the initial average flux. Then, by operating the backwash pump and rinsing with 50 °C water for half an hour, the total flux in the fourth and seventh batches reached the initial level, i.e., the average flux of the first batch was recovered. This phenomenon suggests that salt accumulation on the permeate side reduced the membrane flux. Over the seven batches, the ethyl acetate concentration on the permeate side basically remained constant. It is important to note that the pressure of the cleaning system should not be excessive (lower than 0.01 MPa), or the reverse pressure could destroy the PDMS membrane.

## 4. Conclusions

PV experiments were performed using PDMS/ceramic tubular and PDMS/PEI flat-sheet composite membranes. The obtained SEM–EDX results indicated that salt could permeate the PDMS membrane and accumulate in the support layer. For the two investigated salts, i.e., NaCl and CaCl_2_, the permeate flux decreased in the presence of salt in the feed solution, and the flux greatly decreased when the salt concentration was higher than 7 wt.%. However, the separation factor slightly increased with increasing salt concentrations. The salt type (i.e., NaCl and CaCl_2_) was found to exert almost no influence on the water flux at a given feed-solution salinity. After backflushing, the tubular membrane achieved flux recovery, suggesting that the negative salt deposition effect was reversible. The recovered flux increased with backflushing velocity and water temperature. A backflushing water temperature of 50 °C was high enough to dissolve deposited salt on the membrane’s inner surface at a crossflow velocity of 2 m/s. A maximum recovered flux of 1.78 L/m^2^ h was achieved for t_p_ = 30 min and t_b_ = 5 min, and a longer backflushing duration was unnecessary. The pilot-scale experiment results showed that salt accumulation on the permeate side reduced the membrane fluxes, which could recover their initial values by back-cleaning. This work emphasized the effect of salt on separation performance, which is often overlooked in practical applications. It provides a feasible cleaning process and engineering experience to improve pervaporation technology.

## Figures and Tables

**Figure 1 membranes-12-00404-f001:**
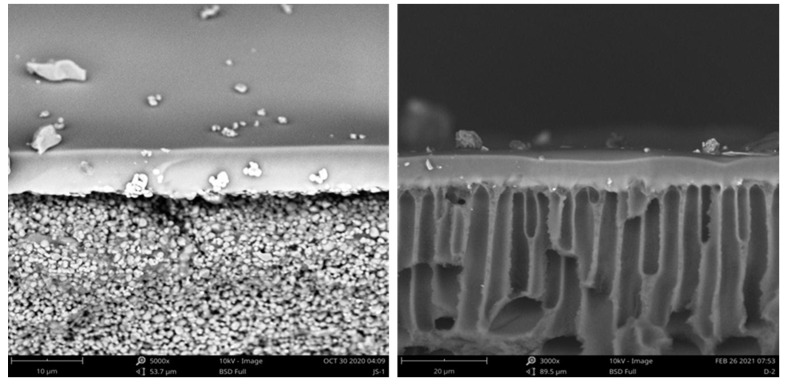
SEM images of the virgin tubular PDMS/Al_2_O_3_/ZrO_2_ composite membrane (**left**) and flat-sheet PDMS/PEI composite membrane (**right**).

**Figure 2 membranes-12-00404-f002:**
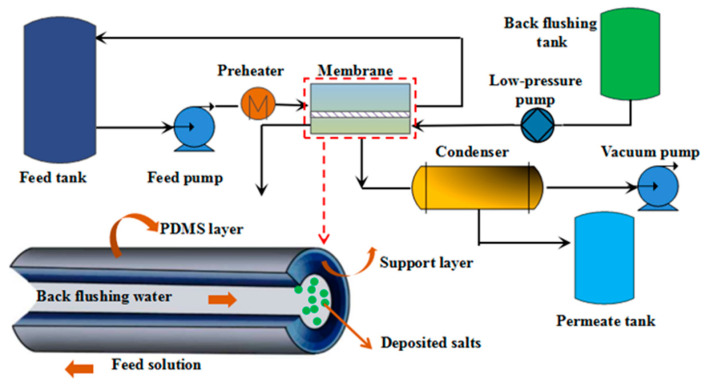
Schematic drawing of the pervaporation and backflushing units.

**Figure 3 membranes-12-00404-f003:**
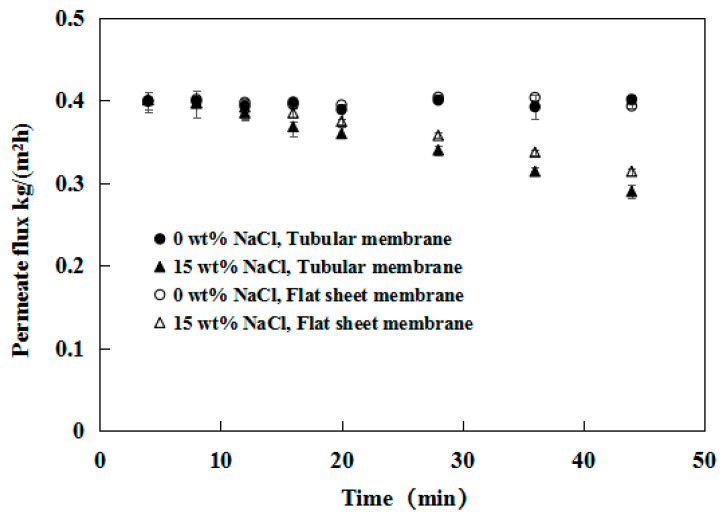
Effect of NaCl concentration and operation time on the water flux in binary NaCl–water solutions. (Feed: NaCl/water solution n).

**Figure 4 membranes-12-00404-f004:**
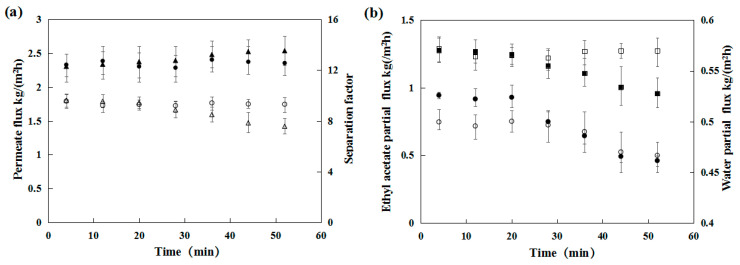
(**a**) Influence of NaCl and operation time on the pervaporation performance in the water–ethyl acetate–NaCl ternary system. (△) Permeate flux (kg/m^2^ h) for 15 wt.% NaCl concentration; (▲) Separation factor for 15 wt.% NaCl concentration; (○) Permeate flux (kg/m^2^ h) for 0 wt.% NaCl concentration: (●) Separation factor for 0 wt.% NaCl concentration; (**b**) Partial flux of Ethyl acetate and water; (●) Water flux (kg/m^2^ h) for 15 wt.% NaCl concentration; (○) Water flux (kg/m^2^ h) for 0 wt.% NaCl concentration; (■) Ethyl acetate flux (kg/m^2^ h) for 15 wt.% NaCl concentration; (□) Ethyl acetate flux (kg/m^2^ h) for 0 wt.% NaCl concentration.

**Figure 5 membranes-12-00404-f005:**
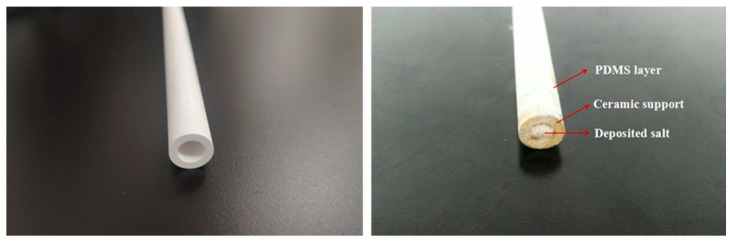
Virgin (**left**) and fouled (**right**) PDMS/ceramic tubular membranes.

**Figure 6 membranes-12-00404-f006:**
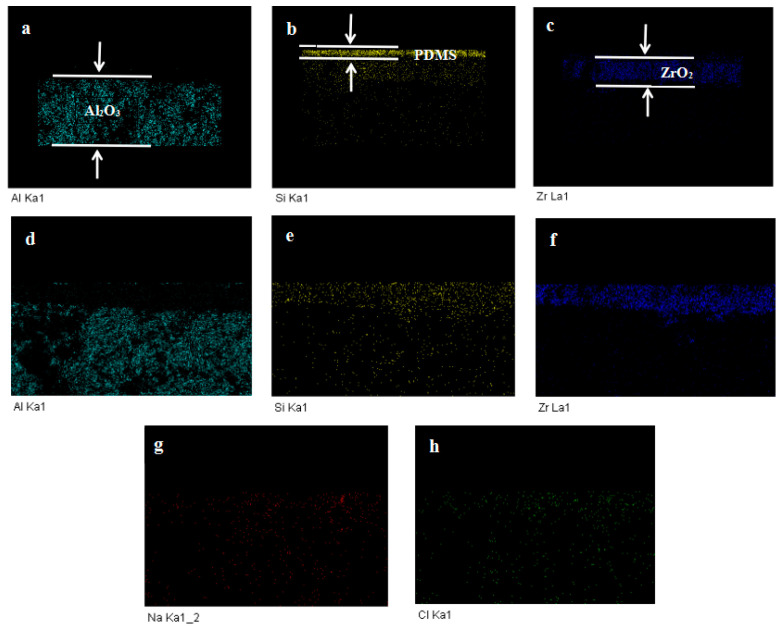
SEM–EDX images of cross-sectional components of the virgin membrane (**a**–**c**) and fouled membrane (**d**–**h**).

**Figure 7 membranes-12-00404-f007:**
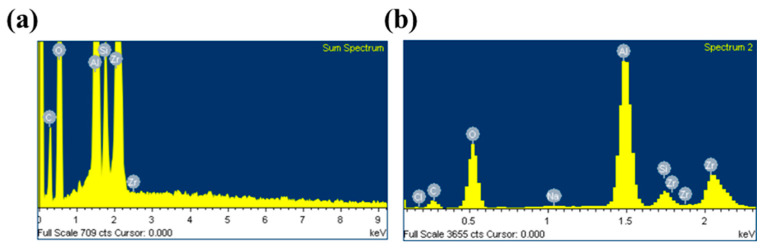
Composition (atomic concentration) of the cross-section: (**a**) virgin membrane; (**b**) fouled membrane.

**Figure 8 membranes-12-00404-f008:**
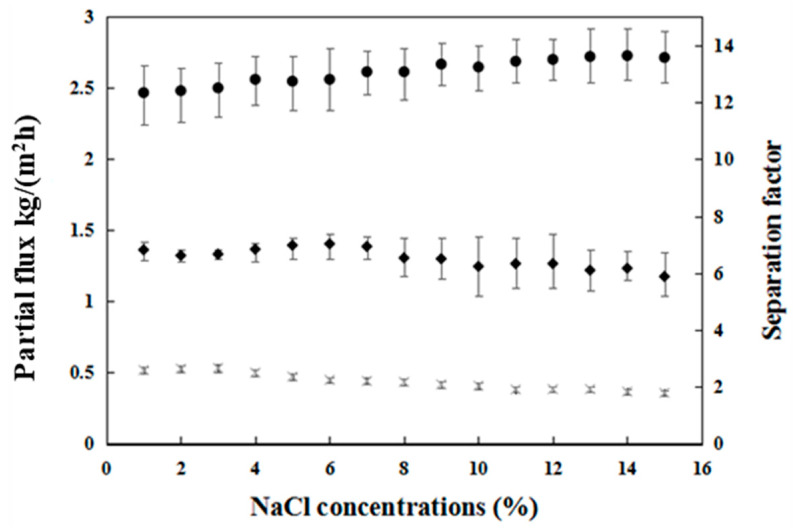
The effect of NaCl concentration on the initial (Δt = 0–10 min) partial permeate flux and separation factor in ethyl acetate/water/NaCl solution: (◆) ethyl acetate flux, (×) water flux, and (●) separation factor.

**Figure 9 membranes-12-00404-f009:**
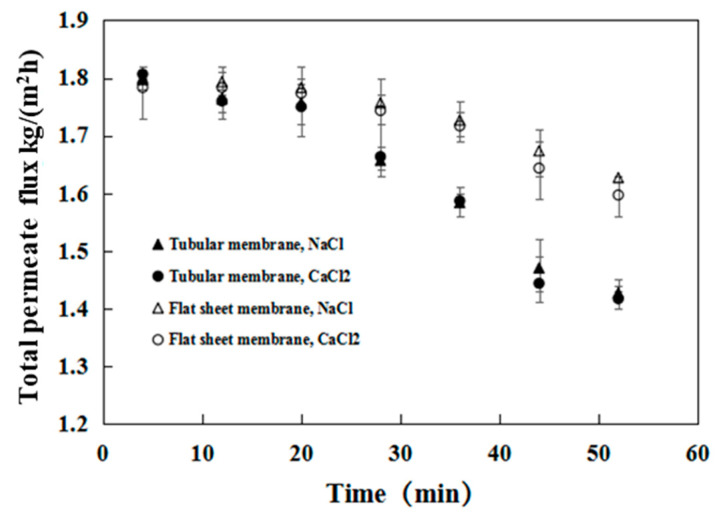
Influence of salt species and operation time on the permeate flux of the tubular and flat-sheet membranes. (Feed: NaCl (15 wt.%)/ethyl acetate (4.8 wt.%)/water and CaCl_2_ (15 wt.%)/ethyl acetate (4.8 wt.%)/water; Operation temperature: 40 °C).

**Figure 10 membranes-12-00404-f010:**
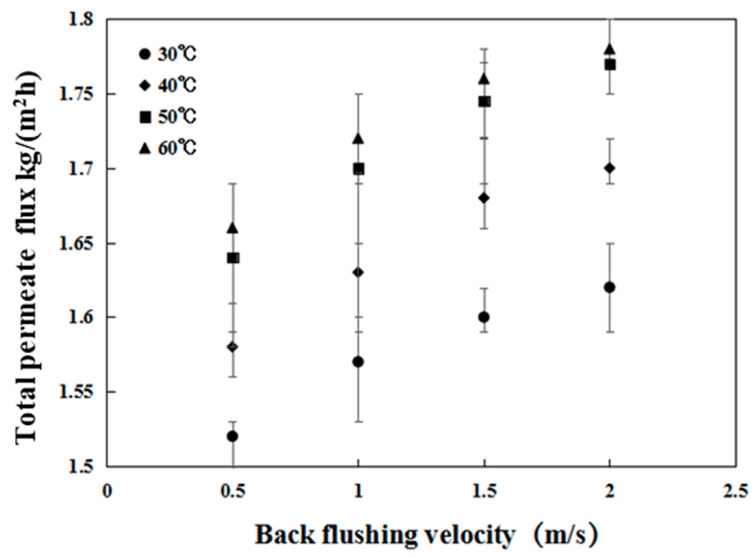
Influence of the backflushing velocity and flushing water temperature on the total permeate flux. (Feed: NaCl (15 wt.%)/ethyl acetate (4.5 wt.%)/water).

**Figure 11 membranes-12-00404-f011:**
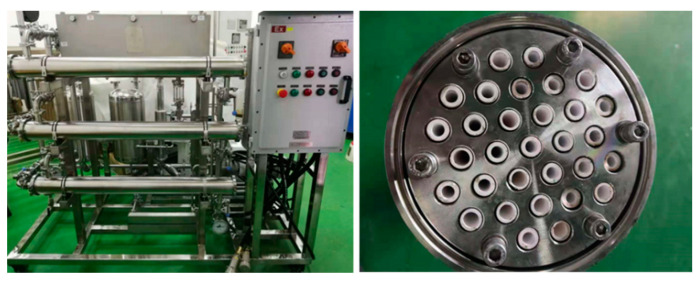
Pilot-scale equipment and membrane module.

**Figure 12 membranes-12-00404-f012:**
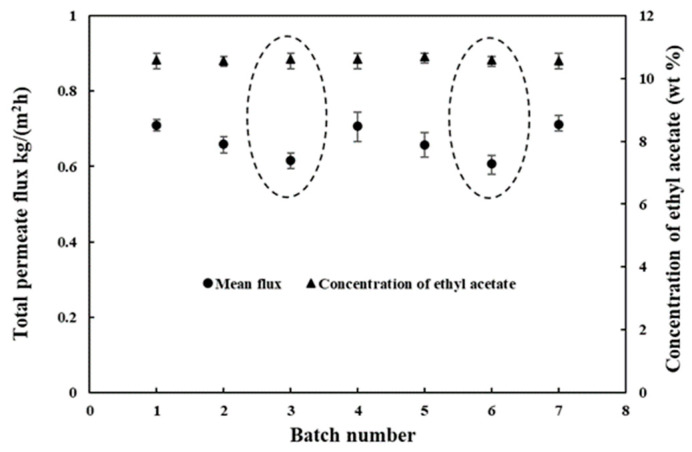
The effect of batches on the separation performance: (▲) Concentration of ethyl acetate (wt.%); (●) Total permeate flux. The data of the third batch is emphasized because the backflushing process was needed to remove the deposited salt. (The composition of feed solution is listed in Table 3; operation temperature is 40 °C.).

**Table 1 membranes-12-00404-t001:** Characteristics of the membranes used in this study.

Membrane	Flat Sheet PV Membrane		Tubular PV Membrane	
Material	Support layer	Activate layer	Support layer	Activate layer
	PEI	PDMS	Al_2_O_3_/ZrO_2_	PDMS
Thickness (um)	30	8	1200	8
Pore size (nm)	50	-	200	-
Membrane area (m^2^)	0.033		0.00785	

**Table 2 membranes-12-00404-t002:** The element composition of the virgin membrane.

Element	Weight%	Atomic%
C K	17.21	27.51
O K	43.15	51.79
Al K	22.02	15.68
Si K	2.78	1.9
Zr L	14.84	3.12
Total	100	

**Table 3 membranes-12-00404-t003:** The element composition of the fouled membrane.

Element	Weight%	Atomic%
C K	15.5	25.64
O K	42.08	52.25
Na K	0.14	0.12
Al K	21.72	15.99
Si K	2.77	1.96
Cl K	0.47	0.27
Zr L	17.31	3.77
Total	100	

**Table 4 membranes-12-00404-t004:** Effect of t_p_ and t_b_ on the recovered flux and the separation factor.

t_p_ (Min)	t_b_ (Min)	Recovered Flux kg/(m^2^ h)	Recovery of the Initial Flux (%)	Separation Factor
No backflushing		1.58	-	12.7
30	5	1.72	98.3	12.1
30	10	1.71	98.9	12.1
60	5	1.65	95.6	12.3
60	10	1.65	95.6	12.4
90	5	1.61	93.3	12.4
90	10	1.60	92.2	12.5

**Table 5 membranes-12-00404-t005:** Parameters of the wastewater quality.

Detection Items	Unit	Detection Value
Turbidity	NTU	7.3
pH		7.6
Total dissolved solids (TDS)	g/L	25.6
K^+^	mg/L	56.2
Na^+^	mg/L	2400
Ca^2+^	mg/L	78.1
Mg^2+^	mg/L	15.2
Chemical oxygen demand (COD)	mg/L	
Ethyl acetate	wt.%	0.65

**Table 6 membranes-12-00404-t006:** Variation in the main components during the pervaporation process.

Time (min)	Ethyl Acetate Concentration (wt.%)		Na^+^ Content (g/L)		EA Flux kg/(m^2^ h)	Water Flux kg/(m^2^ h)	EA Permeance (GPU *)	Separation Factor (EA/Water)
	Retentate	Permeate	Retentate	Permeate				
0	0.65	-	2.4	-	-		-	-
30	0.42	11.32	2.3	0	0.11	0.85	1350	19.5
60	0.27	8.35	2.4	0	0.06	0.71	1228	21.6
90	0.15	3.86	2.4	0	0.02	0.61	719	14.8
120	0.04	0.58	2.5	0	0.003	0.48	147	14.6

* 1 GPU = 3.3928 × 10^−10^ mol/m^2^/s/Pa.

## Data Availability

Data is contained within the article.

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
