# Peer review of "Pollution and Cleaning of PDMS Pervaporation Membranes after Recovering Ethyl Acetate from Aqueous Saline Solutions"

_membranes, 2022, doi:10.3390/membranes12040404_

Round 1

Reviewer 1 Report

The manuscript entitled "Pollution and cleaning of PDMS pervaporation membranes in a salt solution environment" describes the results of lab- and pilot-scale experiments for the separation of ethyl acetate (ΕΑ) from water mixtures through PDMS pervaporation (PV) membranes. The study is focused on the effect of salts’ presence in the Η2Ο/ΕΑ mixtures on the flux and selectivity of the process. The manuscript content is within the journals topics and the content is original to the reviewer’s best knowledge. The manuscript’s objectives are clear, and its novelty is moderate; however, the obtained results are interesting and worth publishing. The methodology employed also seems logical. Nevertheless, the manuscript has a serious flow that needs to be addressed, since it compromises its overall quality. The description of the methodology (i.e. Chapter 2:  2. Materials and method) is very weak since it hardly provides an overview of the experimental procedures. The description lacks the necessary details and the methodological organization that would allow the reader to understand (and possibly replicate) the employed experiments. The authors provide some short descriptions of experimental procedures throughout the manuscript (e.g. Lines 223-225, Lines 261-264, Lines 288-292 etc.), instead providing all these data in Chapter 2. The quality of the manuscript is also compromised by the unprecise tiles of specific Chapters and Sub-chapters, as well as the quality and clearness of the presented plots and of their legends. Finally, the English language is satisfactory, although some points in the manuscript should be clearer or rephrased.

Concerning Chapter 2. the authors are kindly advised to improve its quality, taking into consideration the following issues:

  1. Line 95: The term "salts" is too general. Please specify which salts have been used in this study.
  2. Line 103: Please add "laboratory" before "...operation unit..." given that two experimental setups are used in this study.
  3. Lines 109-110: The authors should describe in detail how did they maintain the EA concentration constant given that it was continuously removed through the PV membrane? The successful application of constant feed parameters is a critical parameter for the experimental soundness of PV experiments.
  4. Line 112: Please provide the feed crossflow velocities for each membrane module used. The term "...turbulent flow..." is not precise enough.
  5. Line 113: Please provide some data concerning the efficiency of the cold trap. Did the authors measure the concentration of EA in the off gases, to confirm that the EA was effectively captured?
  6. Line 123: The authors should also provide a detailed description of the experimental set-up for the pilot-scale tests. The authors provide a short description in Chapter 3.3 (together with some pictures), which may be used as a basis for a detailed description of the pilot unit provided in Chapter 2.2.
  7. After the end of Chapter 2.2, the authors should add another Chapter where they will provide a detailed description of the experimental procedures and tests. The authors provide fragmented descriptions of some experimental procedures throughout the manuscript. These parts should be systematically provided in a separate Chapter.
  8. The authors also fail to provide any data on whether there were any replicates in their experiments.
  9. Chapter 2.3 is also incomplete. The authors in the manuscript provide the following terms: "constant flux" and "steady-state flux". What are the definitions of these two different terms? How did they calculate the "constant flux" and the "steady-state flux"?
  10. The authors should provide details on the analytical procedures that they used to for example determine the concentration of EA, the conductivity of the permeate, the operating parameters of the SEM measurements etc.

Apart from the aforementioned issue concerning the description of the experimental procedure, the authors are kindly asked to take into consideration the following comments:

  1. The title is too general and does not provide an accurate description of the manuscript's content, e.g. "Pollution and cleaning of PDMS pervaporation membranes for ethyl acetate recovery from saline aqueous solutions".
  2. Line 49: The phrase "...inorganic salts are usually contained in organic solvents..." is not very clear.
  3. Line 54: Please consider rephrasing "...side remain controversial subjects." to ""...side is a controversial subject.". Please, also consider deleting the paragraph change between Lines 54 and 55.
  4. Line 64: The term "penetrant" is not accurate; probably "permeate" is a better term.
  5. Line 65: Please add "other" before "... researchers have reported...".
  6. Lines 66-69: This reference is irrelevant to the manuscript's content since it refers to a membrane distillation process employing porous membranes. PV comprises non-porous membranes and is fundamentally different from the MD process. Please consider removing this reference as irrelevant.
  7. Line 76: The authors could also try to provide some explanations on why is there such a significant difference/controverse between the different studies concerning salt permeation.
  8. Lines 140-146: Please consider replacing the term "permeate flux" with "water flux" given that the permeate is merely water in these experiments.
  9. Line 148: Please consider replacing the word "performance" with "flux".
  10. Line 151: This argument is not well-supported. The differences in the presented results in Fig. 3 are very small and could be easily attributed to the analytical errors in measurements. Did the authors identify the experimental error and accuracy of their measurements?
  11. Lines 152-153: This statement is also not supported from the presented results. The authors do not provide any comparison between the salt permeation for the two membranes employed. Thus, how did they reach to this conclusion?
  12. 3 legend: Please rewrite as follows: "Figure 3. Effect of NaCl on the water flux in binary NaCl-water solutions."
  13. 4, 8 and 12: You may remove the arrows that specify the y-axis, by better defining the symbols in the plot. For example, in Fig.4, for the open circles you may define it as "Permeate flux (kg/m2 h) for 0 wt % NaCl concentration" instead of "0 wt% NaCl".
  14. Lines 170-172: Please rephrase the sentence "Although the...through the membrane" to "Although the permeate conductivity was very low, i.e. 4.1 μS/cm, the salts could obviously permeate through the membrane".
  15. Line 176: Please consider rephrasing the phrase "...could not freely move and..." as follows: "...they subsequently crystallized and...".
  16. Lines 185-186: The authors provide strong evidence that the PDMS polymer is present in the support layer of the tubular membranes. Can we consider that EA dissolves or swells the PDMS polymer? Please comment (see also https://doi.org/10.1021/ac0346712).
  17. 6: The authors should provide a scale bar in all SEM pictures. To better clarify the SEM images, it would be better to provide the same scale in all images and indicate (if possible) the borders of the membrane's top and bottom edges.
  18. 8: The authors should specify to which time point do the presented data refer to? Are they average values? The Fig. legend should also be revised as: "Figure 8. The effect of NaCl concentration on the permeate flux and separation factor in ethyl acetate/water/NaCl mixtures.
  19. Lines 206-207: The sentence is not clear; please consider rephrasing.
  20. Line 219: "Since the activity of ethyl acetate increases by adding NaCl to the solution..." the authors should provide some references to support this statement.
  21. Line 227: The term "impermeable salts" is contradictory to the general outcome of this study that NaCl do permeate through the PV membrane. The authors should preferably delete the word "impermeable".
  22. Lines 250-251: Please rephrase the sentence accordingly: "As discussed above, there was a negative effect on the PV separation performance at high salt concentrations in the feed solution."
  23. Line 280: Please consider replacing "...relatively low..." with "...lower...".
  24. Lines 281-285: Please consider the following syntax: "However, as the backflushing temperature was increased from 50°C to 60°C, the cross-flow velocity did not result to a clear improvement since the steady-state flux slightly increased at all cross-flow velocities. This indicates that a temperature around 50°C is sufficiently high to dissolve any deposited salts on the membrane inner surface, and the further increase in back flushing water temperature does not significantly influence the steady-state flux."
  25. Line 297: Please consider rephrasing "...there occurred a flux decline of 8.3%..." to "a flux decline of 8.3% occurred"
  26. Lines 300-301: The sentence "The obtained...no influence" is unclear; please consider rephrasing it.
  27. Line 339: The phrase "...and could...irrigation." is unclear.
  28. Lines 340-343: In these experiments, given that the feed concentration is not constant, authors should calculate the permeance, i.e. permeate flux/driving force and compare the membrane performance at the different time points.
  29. Table 4: The column "Permeate flux" refers to the flux of EA or the EA/water mixture? In the first case, the authors can they provide data on the permeate flux of the water as well and calculate the PV membrane selectivity?
  30. Lines 349-351: The description of the cleaning procedure is too simplified; e.g. what was the duration, pressure, or flow of the air flushing?
  31. Line 352: The word "gradually" has a link to an unknown site. What is the purpose for this link?
  32. Line 366: Please consider this rephrasing "For the two investigated salts, i.e., NaCl and CaCl2,"
  33. Line 370: Please replace "slightly influence" with "hardly result to any influence on"
  34. Line 374: Please delete "any".
  35. Lines 381-385: The authors should fill these fields.

Author Response

We thank the reviewer's comments very much. Please find the point-to-point response in the attached file. 

Reviewer 2 Report

The manuscript presents an interesting work based on accurate results, and supported by an interesting discussion. Anyway, in my opinion the introduction and discussion could be improved just by updating the references used, since most of them are before 2010, and just a few of 2013 and 2014. I encourage you to update the references used in order to improve the quality of the manuscript.

Author Response

(The authors gave the same response as above.)

Reviewer 3 Report

The submitted manuscript the dealing with the removal of volatile organic compounds from aqueous solutions containing inorganic salts. by using pervaporation. From this point of view, the manuscript fits well the profile of MDPI Membranes. However the way of the presentation of results is unfortunately unacceptable for the journal. The most important inaccuracies disqualifying the work are presented in section 2 - Authors claimed that the membrane thickness was 5 μm (line 90) whereas in Table 1 the value 8 μm is given. Moreover, in line 114 Authors stated that the cold traps were changed every 30 minutes but in the all figures the whole experiment lasted 45 minutes (e.g. Fig. 3) and Authors present around 10 experimental points. During pervaporation process, at least 60 minutes is needed to reach so called stationary state. Therefore, I have to recommend the rejection of the manuscript without an option for the resubmission.

Author Response

(The authors gave the same response as above.)

Round 2

Reviewer 1 Report

The authors have successfully addressed most of the reviewer’s concern and the quality of the revised manuscript has been greatly improved. There are still some points that the authors are advised to clarify/address, to further upgrade their manuscript.

More specifically (the line’s numbers refer to the revised manuscript):

  1. Line 50: The sentence could be rephrased as follows: "However, inorganic salts are also usually present in wastewater containing organic pollutants, limiting..."
  2. Line 113: Please consider rephrasing "...dissolve immediately to maintain..." as "...dissolve, maintaining..."
  3. Line 146: Given that the experiments were performed in triplicate, please provide the Standard Deviation (SD) of the measurements in Figures 3,4,8,9 and 12.
  4. Line 154: Please provide a definition of the term "steady-state flux" which is used in the presentation of the results.
  5. Lines 198-200: Given that no error bars are provided in Fig.3, the reviewer has strong doubts on whether the presumably greater flux decline of the ceramic membrane is statistically significant. Moreover, did the authors quantify the salt permeance in the two membranes? How did the declare that "...different salt pollution degrees of the support layers." took place?
  6. Figure 8: Given that the presented results refer to a specific time point the Figure legend should be updated as follows: "The effect of NaCl concentration on the initial (Δt = 0-10min) permeate flux and separation factor in ethyl acetate/water/NaCl mixtures. (▲) Permeate flux (kg/m2 h), (●) Separation factor.
  7. Line 262: Please consider rephrasing the phrase "...salts diffusing in water first block channels facilitating..." to "...diffusing salts seems to block channels that facilitate..."
  8. Table 4: The term "Total permeate flux" may be also confusing to the reader since he/she may consider it as the total in terms of time (i.e. an integrated value). I propose to keep the initial term "Permeate flux" and in line 384 add an explanation as follows: "...permeate flux (of the EA/water mixture) gradually decreased from...". Moreover, in EA permeance please replace GPU units with the SI units (mol/(m2 s Pa).

Author Response

Please see the point-by-point response in the attached file. 

Reviewer 3 Report

I have read the revised version but still I am confused, what and how was measured. Unfortunately Authors removed from the experimental section information about the procedure of data gathering. There is no information about the period needed for the reaching the stationary state, how many cold traps were used (1 or 2 in parallel). This is basic information without which the evaluation of manuscript is impossible.
The title of section 2.4 in really misleading - two equations is not a theory. Moreover, Authors wrongly use terminology. In pervaporation selectivity coefficient (α) and separation factor (β) are two absolutely different terms. The first is related with the ratio of permeances (Pi/Pj) and the second is defined by Eq. (2) - line 181/182. 
Results presented in Fig. 3 suggest that the permeate trap was exchange every few minutes and the total experiment lasted around 45 minutes. Once more the question - what about the stationary state? Y axes - it should be stated that data shows permeate water flux. 
Fig. 4 - which permeate flux was shown? If total it is again the wrong approach: both permeate fluxes MUST be shown, i.e. water + EtAc, as both are changing with time. The legend put in Figure is much more informative than the extended figure caption.
Fig. 8 - again partial fluxes MUST be shown. The discussion given in lines 257-276 can be hardly related with data presented in this figure (e.g. line 259 - a drop in the total flux is observed after 50 min of permeation). With increasing of the salt concentration the driving force is decreasing and it is not related with changes of density or viscosity (lines 266-268).
The same comments - which flux is shown in Fig. 9? If there are 2 liquid components, both fluxes should be shown.
Fig. 10 - unclear results with regard of the flux and temperature. Which temperature was changed? Back flushed medium (water) with constant feed temperature?
Fig. 12 - in the plot the point cannot be connected and moreover the modul rinsing after 3rd batch should be indicated.
What is the value of the salt transport across NaCl? It could be estimated. The observed transport can be caused also by a pinholes in the PDMS layer. Did Authors make some additional experiments  to exclude such hypothesis (e.g. diffusional tests).
Unfortunately, after reading and analysis the revised version I have again recommend the rejection of the manuscript without an option for the resubmission. There are too much questions which left unanswered.

Author Response

(The authors gave the same response as above.)

Round 3

Reviewer 3 Report

I have noticed that the manuscript was slightly improved but still the information given, regarding the mode of permeate collection, is really contradictory: in lines 149-151 Authors claimed that traps were changed every 30 minutes but in figures presenting the collected data this time is equal to less than 10 minutes. Moreover, presenting the total flux values in the plots is not needed (Fig. 8 - figure caption is incomplete, remove total flux; Fig 9 - figure caption is incomplete - which flux is presented, what was the feed mixture; units - should be presented as kg m-2 h-1 or kg/(m2 h); Fig. 10 - figure caption and description of Y axis, units; Tab. 2 - recovered flux - the format of units is wrong, moreover is given in L  m-2 h-1, the units should be unified in the text; Table 4 - units for the permeate flux /NB - both permeate fluxes MUST be shown/ and permeance of EA are wrong).

Author Response

(The authors gave the same response as above.)
